# Estimation of stillbirths attributable to ambient fine particles in 137 countries

Tao Xue [1,8] ✉, Mingkun Tong[1,8], Jiajianghui Li[1], Ruohan Wang[1], Tianjia Guan[2], Jiwei Li[3], Pengfei Li[1,4,5], Hengyi Liu[1], Hong Lu[1], Yanshun Li[6] & Tong Zhu [7]

Gestational exposure to ambient fine particles (PM$_{2.5}$) increases the risk of stillbirth, but the related disease burden is unknown, particularly in low- and middle-income countries (LMICs). We combine state-of-the-art estimates on stillbirths, and multiple exposure–response functions obtained from previous meta-analyses or derived by a self-matched case-control study in 54 LMICs. 13,870 stillbirths and 32,449 livebirths are extracted from 113 geocoded surveys from the Demographic and Health Surveys. Each stillbirth is compared to livebirth(s) of the same mother using a conditional logit regression. We find that 10-μg/m$^3$ increase of PM$_{2.5}$ is associated with an 11.0% (95% confidence interval [CI] 6.4, 15.7) increase in the risk of stillbirth, and the association is significantly enhanced by maternal age. Based on age-specific nonlinear PM$_{2.5}$–stillbirth curves, we evaluate the PM$_{2.5}$-related stillbirths in 137 countries. In 2015, of 2.09 (95% CI: 1.98, 2.20) million stillbirths, 0.83 (0.54, 1.08) million or 39.7% (26.1, 50.8) are attributable to PM$_{2.5}$ exposure exceeding the reference level of 10 μg/m$^3$. In LMICs, preventing pregnant women from being exposed to PM$_{2.5}$ can improve maternal health.

The United Nations (UN) calls the global burden of stillbirths a neglected tragedy[1]. The UN Inter-agency Group for Child Mortality Estimation (UN IGME) found that there were 2.0 million stillbirths (90% confidence interval [CI]: 1.9, 2.2) in 2019 globally, and recognized that progress in stillbirth prevention had slowed, particularly in low- and middle-income countries (LMICs), such as those in sub-Saharan Africa[2]. Most (98%) stillbirths are estimated to occur in LMICs[3]. However, this issue has received little attention. For instance, stillbirth was not included in the Millennium Development Goals[4] or tracked by the Global Burden of Disease (GBD) study[5], both of which counted live-births only. Stillbirth is linked to disease burden such as bleeding or infection[6], secondary infertility[6,7], and psychological sequelae (grief, anxiety, posttraumatic stress) affecting the whole family[8,9]. In addition,

the economic costs of stillbirth (including health-care costs and loss caused by the incapacity to work) affect individuals, healthcare systems, and society[6]. Therefore, stillbirth intervention could promote maternal health and sex equality.

Preventing stillbirth depends on a comprehensive understanding of the underlying risk factors. Recent studies have shown an epidemiological association between gestational exposure to fine particulate matter (PM$_{2.5}$) and stillbirth[10–14]. Two meta-analyses of the association between stillbirth and PM$_{2.5}$ relied on evidence found before 2020. Zhang et al.[15] reported a pooled OR of 1.103 (95% CI 1.074, 1.131) per 10 μg/m$^3$ increment in PM$_{2.5}$ during pregnancy based on seven independent studies; Xie et al.[11] reported a value of 1.15 (95% CI 1.07, 1.25) based on six of those seven studies. Due to the ubiquity of

[1]Institute of Reproductive and Child Health / National Health Commission Key Laboratory of Reproductive Health and Department of Epidemiology and Biostatistics, School of Public Health, Peking University Health Science Centre, Beijing, China. [2]Department of Health Policy, School of Health Policy and Management, Chinese Academy of Medical Sciences & Peking Union Medical College, Beijing, China. [3]School of Computer Science, Zhejiang University, Hangzhou, China. [4]Advanced Institute of Information Technology, Peking University, Hangzhou, China. [5]National Institute of Health Data Science, Peking University, Beijing, China. [6]Department of Energy, Environmental & Chemical Engineering, Washington University in St. Louis, St. Louis, MO, USA. [7]College of Environmental Sciences and Engineering, Peking University, Beijing, China. [8]These authors contributed equally: Tao Xue, Mingkun Tong. ✉e-mail: txue@hsc.pku.edu.cn

$PM_{2.5}$ pollution, it may be a major contributor to the global burden of stillbirth.

How $PM_{2.5}$ contributes to the global burden of stillbirth is unknown because of the following knowledge gaps. First, there is significant heterogeneity in $PM_{2.5}$–stillbirth associations between studies[11,13]. A major reason for this is differential susceptibility according to demographic characteristics. For instance, the most mature exposure–response curves[16], developed to assess the cardiovascular mortality risk for $PM_{2.5}$, are stratified by age group. Similarly, maternal age can significantly modify the $PM_{2.5}$–stillbirth association[10,12]. Therefore, a representative and age-specific exposure–response curve between $PM_{2.5}$ and stillbirth is needed. Second, a few key inputs (e.g., the baseline risk of stillbirth and high-resolution spatial distribution of the population at risk) into the risk assessment of stillbirth became available only recently. The UN IGME developed the first state-of-the-art estimates of the total number of stillbirths for 195 countries[2], and WorldPop generated gridded maps of pregnancies (including livebirths and stillbirths) with a spatial resolution of $1 \times 1$ km for 161 countries or regions[17], covering Africa, Latin America and the Caribbean, which have high stillbirth rates.

We have developed a self-matched case-control method to evaluate the association between $PM_{2.5}$ and stillbirths in Africa[10] and South Asia[14]. The method has been recommended as a cost-effective approach to developing an exposure–response curve for $PM_{2.5}$ and stillbirth[18] from large-population data in LMICs. This study aims to present an assessment study to quantify the burden of $PM_{2.5}$-related stillbirth, using exposure-response curves derived from our approach or other meta-analyses. Although establishing exposure-response curves was not within our major study aim, this study increased confidence on estimates by enlarging sample size, and specified the curves by age groups, compared to our previous analyses. Combining the curves with state-of-the-art estimates on the population at risk, $PM_{2.5}$ concentration, and baseline risk, we evaluated the number of stillbirths attributable to $PM_{2.5}$ exposure in 137 countries (Supplementary Table 1) from 2000 to 2019.

## Results

### Exposure–response curve

To establish the exposure–response curves for risk assessment, we analyzed 46,319 cases of gestation linked to 13,870 mothers from 1998 to 2016. The mean maternal age in the control group was 24.97 years with a standard deviation (SD) of 6.02 years, younger than the stillbirth group (mean 26.56 years; SD 7.02 years). The mean length of intervals between stillbirth and livebirth was 3.81 (SD = 2.45) years (Supplementary Table 2). Nightlight, as an indicator of developmental level, was 8.29 digit-number (DN) (SD = 16.14 DN) for the case group, higher than that for the control group (mean = 7.07 DN; SD = 14.99 DN; $p$ value $<2\times10^{-16}$ for a paired test). This suggests that more stillbirths occur in more developed regions. The controls had a lower level of gestational exposure to $PM_{2.5}$ (mean = 40.34 $\mu g/m^3$, SD = 22.20 $\mu g/m^3$), compared to stillbirth cases (mean = 40.96 $\mu g/m^3$; SD = 23.04 $\mu g/m^3$; $p$ value $<2\times10^{-16}$ for a paired test). Country-specific distributions for the environmental variables are shown in Supplementary Fig. 1. The population characteristics for stillbirth and the secondary outcomes analyzed in this study are summarized in Table 1. The spatial distributions of the surveyed samples from the 54 LMICs are shown in Fig. 1a, along with the exposure level of $PM_{2.5}$ in 2015.

We examined the linear association between $PM_{2.5}$ and stillbirth. We found a robust association between $PM_{2.5}$ levels and stillbirths; the level of significance was not sensitive to adjustments of different covariates (Supplementary Fig. 2), and the association was not heterogenous between most subpopulation groups except between different maternal ages (Supplementary Fig. 3). According to the fully adjusted model, each 10 $\mu g/m^3$ increment in $PM_{2.5}$ was associated with a 11.0% (95% CI: 6.4, 15.7) increased risk of stillbirth. Regarding

secondary outcomes, the association was estimated to be −1.4% (95% CI: −5.3, 2.7), 2.3% (95% CI: 1.5, 3.2) or 0.5% (95% CI: −0.4, 1.4) for early stillbirth, miscarriage, or pregnancy loss, respectively (Supplementary Fig. 2). Among all subtypes of pregnancy loss, stillbirth was most strongly associated with $PM_{2.5}$ exposure. In addition, advanced maternal age significantly enhanced the $PM_{2.5}$–stillbirth association ($P = 0.032$). For more details on subpopulation indicators and subpopulation-specific associations, please see Supplementary Table 2 and Supplementary Fig. 3.

The linear models acted as the preliminary explorations for the development of nonlinear exposure–response curves. The findings suggested that the nonlinear curves should be stratified by maternal age. Using the fully adjusted model, we derived the nonlinear $PM_{2.5}$–stillbirth associations with or without age-stratification (Fig. 2, Supplementary Fig. 4 and Supplementary Fig. 5). Generally speaking, the nonlinear models showed a sublinear curvature of the exposure–response relationships for both stillbirth and the secondary outcomes (Supplementary Fig. 4, and Supplementary Fig. 5). Consistent with the subpopulation-specific results estimated using linear models, the nonlinear curves generally showed a higher risk for mothers with more advanced maternal age, particularly those >34 years old. However, at different exposure levels, the significance of the modifying effect was different. At high exposure levels, the modification by age tended to be apparent (Fig. 2 and Supplementary Fig. 5). Therefore, a combination of the nonlinear curvature and the age's modification could partially explain the heterogeneity between the country-specific linear associations between $PM_{2.5}$ and stillbirth (Supplementary Fig. 6). In addition, uncertainties embedded in the pointwise estimates along the exposure–response curves were nonnegligible. For instance, according to the pointwise estimates, compared to the meta-analysis-based log-linear association reported by Xie et al.[11], the sublinear curves showed higher risk at a low exposure level or lower risk at a high exposure level (Fig. 2). However, given the uncertainties, our exposure–response curves are comparable to those in prior meta-analyses (Fig. 2).

### Exposure assessment

Among the 137 countries, in 2015, the pregnancy-number-weighted average of $PM_{2.5}$ was 43.24 $\mu g/m^3$ for all populations at risk. Regarding age-specific subpopulations, the average of $PM_{2.5}$ was 40.09, 44.8, 41.84, and 40.36 $\mu g/m^3$ for pregnancies at maternal ages of <20, 20–29, 30–34, and >34 years. These results suggest that the exposure level among women at peak reproductive age (20–34 years) was higher than that in the adolescent pregnancy (<20 years old) and geriatric pregnancy (>34 years old) groups. Across locations where $PM_{2.5}$ concentration was from 10 to 100 $\mu g/m^3$, the proportion of pregnant women 20–29 years old was positively correlated with exposure level (Fig. 3). This correlation can in part explain the differences in age-specific $PM_{2.5}$ averages. Population distributions by $PM_{2.5}$ for different countries and regions are shown in Supplementary Fig. 7 and Supplementary Fig. 8, respectively.

We also calculated the number of pregnancies exposed to $PM_{2.5}$ above the WHO reference level. Of the pregnancies in the study domain, 99.96%, 98.87%, 93.51%, 73.16%, and 53.69% were beyond the WHO AQG (5 $\mu g/m^3$), IT4 (10 $\mu g/m^3$), IT3 (15 $\mu g/m^3$), IT2 (25 $\mu g/m^3$), and IT1 (35 $\mu g/m^3$), respectively (Fig. 3). For the cleanest of the 137 countries, the Bahamas (pregnancy-number-weighted average of $PM_{2.5} = 5.58 \mu g/m^3$), 82.90% of pregnancies were above the AQG. Therefore, the AQG might be too rigorous, and thus we selected IT4 as our major referent level for minimum risk ($C_0$) in the assessment model (Eq. 4). For instance, in the dataset used to develop the exposure–response curves, no sample was exposed to $PM_{2.5} < 5 \mu g/m^3$ (Fig. 2). Effects estimated based on the $PM_{2.5}$–stillbirth curves for such low levels of exposure were extrapolations and thus unreliable. Country-level assessments of exposure are shown in Supplementary Table 3.

**Table 1 | Population characteristics for the dataset used to establish the exposure–response curves between PM$_{2.5}$ and stillbirth or other similar outcomes**

| Variable | Group | Outcomes and their matched controls | | | |
|---|---|---|---|---|---|
| | | Stillbirth | Early stillbirth | Miscarriage | Pregnancy loss |
| Categorical variables | | Number (percentage) | | | |
| Total | | 46,319 (100%) | 32,665 (100%) | 279,147 (100%) | 358,131 (100%) |
| Pregnancy loss | Control | 32,449 (70.1%) | 22,882 (70.1%) | 193,599 (69.4%) | 248,930 (69.5%) |
| | Case | 13,870 (29.9%) | 9,783 (29.9%) | 85,548 (30.6%) | 109,201 (30.5%) |
| Number of matched controls | 1 | 10,162 (21.9%) | 6,864 (21.0%) | 61,494 (22.0%) | 78,520 (21.9%) |
| | 2 | 11,556 (24.9%) | 8,400 (25.7%) | 78,573 (28.1%) | 98,529 (27.5%) |
| | 3 | 8,988 (19.4%) | 6,924 (21.2%) | 58,152 (20.8%) | 74,064 (20.7%) |
| | 4 | 7,015 (15.1%) | 4,930 (15.1%) | 38,315 (13.7%) | 50,260 (14.0%) |
| | 5+ | 8,598 (18.6%) | 5,547 (17.0%) | 42,613 (15.3%) | 56,758 (15.8%) |
| Maternal age (years) | <20 | 9,712 (21.0%) | 7,081 (21.7%) | 51,841 (18.6%) | 68,634 (19.2%) |
| | 20–29 | 25,699 (55.5%) | 17,548 (53.7%) | 157,077 (56.3%) | 200,324 (55.9%) |
| | 30–34 | 6,638 (14.3%) | 4,667 (14.3%) | 41,342 (14.8%) | 52,647 (14.7%) |
| | > 34 | 4,270 (9.2%) | 3,369 (10.3%) | 28,887 (10.3%) | 36,526 (10.2%) |
| Level of nightlight (NTL, Digit-Number) | Low (NTL ≤ 4) | 29,464 (63.6%) | 18,947 (58.0%) | 130,672 (46.8%) | 179,083 (50.0%) |
| | Middle (4 < NTL ≤ 20.5) | 10,310 (22.3%) | 7,585 (23.2%) | 73,896(26.5%) | 91,791 (25.6%) |
| | High (NTL > 20.5) | 6,545 (14.1%) | 6,133 (18.8%) | 74,579 (26.7%) | 87,257 (24.4%) |
| Parity | Nulliparous | 13,175(28.4%) | 8,634 (26.4%) | 75,080 (26.9%) | 96,889 (27.1%) |
| | Multiparous | 33,144 (71.6%) | 24,031 (73.6%) | 204,067 (73.1%) | 261,242 (72.9%) |
| Continuous variable | | Mean (Standard deviation, interquartile range) | | | |
| Maternal age (years) | Total | 25.45 (6.38, 20.58 ~ 29.58) | 25.56 (6.62, 20.50 ~ 29.83) | 25.90 (6.43, 21.00 ~ 30.00) | 25.81 (6.44, 20.92 ~ 29.92) |
| | Control | 24.97 (6.02, 20.42 ~ 28.83) | 24.92 (6.23, 20.17 ~ 29.00) | 25.01 (5.99, 20.50 ~ 28.83) | 25.00 (6.02, 20.42 ~ 28.83) |
| | Case | 26.56 (7.02, 21.08 ~ 31.50) | 27.05 (7.22, 21.50 ~ 32.00) | 27.90 (6.92, 22.58 ~ 32.58) | 27.66 (6.98, 22.33 ~ 32.42) |
| Temperature (°C) | Total | 23.67 (4.68, 21.48 ~ 26.76) | 23.63 (5.06, 21.26 ~ 26.84) | 23.20 (6.24, 20.16 ~ 27.14) | 23.30 (5.96, 20.49 ~ 27.03) |
| | Control | 23.74 (4.70, 21.54 ~ 26.81) | 23.69 (5.09, 21.32 ~ 26.87) | 23.22 (6.23, 20.24 ~ 27.11) | 23.33 (5.95, 20.57 ~ 27.03) |
| | Case | 23.53 (4.64, 21.37 ~ 26.60) | 23.50 (4.99, 21.12 ~ 26.77) | 23.14 (6.26, 19.94 ~ 27.20) | 23.22 (5.97, 20.31 ~ 27.05) |
| Nightlight (Digit-Number) | Total | 7.44 (15.35, 0.00 ~ 6.40) | 9.70 (17.51, 0.00 ~ 9.00) | 13.69 (20.12, 0.00 ~ 19.00) | 12.52 (19.47, 0.00 ~ 16.00) |
| | Control | 7.07 (14.99, 0.00 ~ 5.75) | 9.11 (17.01, 0.00 ~ 8.33) | 12.93 (19.67, 0.00 ~ 17.00) | 11.81 (19.01, 0.00 ~ 14.00) |
| | Case | 8.29 (16.14, 0.00 ~ 7.20) | 11.06 (18.54, 0.00 ~ 12.00) | 15.42 (21.01, 0.00 ~ 25.00) | 14.12 (20.40, 0.00 ~ 21.00) |
| PM$_{2.5}$ (μg/m$^3$) | Total | 40.53 (22.46, 24.40 ~ 49.65) | 38.37 (24.53, 22.01 ~ 46.40) | 40.01 (29.54, 21.20 ~ 48.20) | 39.93 (28.30, 21.65 ~ 48.20) |
| | Control | 40.34 (22.20, 24.41 ~ 49.38) | 38.07 (24.01, 22.07 ~ 46.22) | 39.57 (29.25, 21.00 ~ 47.70) | 39.53 (27.98, 21.53 ~ 47.78) |
| | Case | 40.96 (23.04, 24.35 ~ 50.22) | 39.07 (25.69, 21.87 ~ 46.89) | 41.01 (30.17, 21.57 ~ 49.27) | 40.83 (28.98, 21.92 ~ 49.17) |

**Risk assessment**

According to UN IGME estimates[2], globally, there were 2,131,914 stillbirths in 2015. Of them, 98.03% (2,089,918) were from the 137 countries in our assessment. In those countries, on average, 45.51% (95% CI: 29.24, 58.07), 39.66% (95% CI: 26.07, 50.85), 33.50% (95% CI: 22.77, 42.70), 22.05% (95% CI: 15.55, 27.71), and 13.45% (95% CI: 8.60, 17.68) of those stillbirths were attributable to gestational exposure to PM$_{2.5}$ exceeding the WHO AQG, IT4, IT3, IT2, and IT1, respectively. The attributable fraction indicated a total number of 0.95 (95% CI: 0.61, 1.23) million, 0.83 (95% CI: 0.54, 1.08) million, 0.70 (95% CI: 0.48, 0.90) million, 0.46 (95% CI: 0.32, 0.58) million, and 0.28 (95% CI: 0.18, 0.37) million PM$_{2.5}$-associated stillbirths in 2015, respectively. Considering the uncertainties, our results are not sensitive to use of different exposure–response curves. Referring to the main level of minimum

risk (i.e., IT4), the number of stillbirths attributable to PM$_{2.5}$ was estimated to be 0.81 (95% CI: 0.41, 1.12) million, 0.54 (95% CI: 0.38, 0.68) million, and 0.70 (95% CI: 0.30, 1.02) million according to the curve derived from the meta-analysis conducted by Xie et al.[11], the meta-analysis by Zhang et al.[15], or our all-ages model, respectively (Supplementary Fig. 9). Those estimates are comparable to the result from our main model: 0.83 (95% CI: 0.54, 1.08) (i.e., the age-specific curves).

The point-estimates from various PM$_{2.5}$–stillbirth curves were affected by the referent exposure level of minimum risk used. When using a strict referent (e.g., AQG or IT4), point-estimates from the sublinear curves were larger than those from log-linear curves based on previous meta-analyses. When using a loose referent (e.g., IT1), among the four curves, the log-linear curve reported by Xie et al.[11] produced the largest point-estimates (Supplementary Fig. 9). The

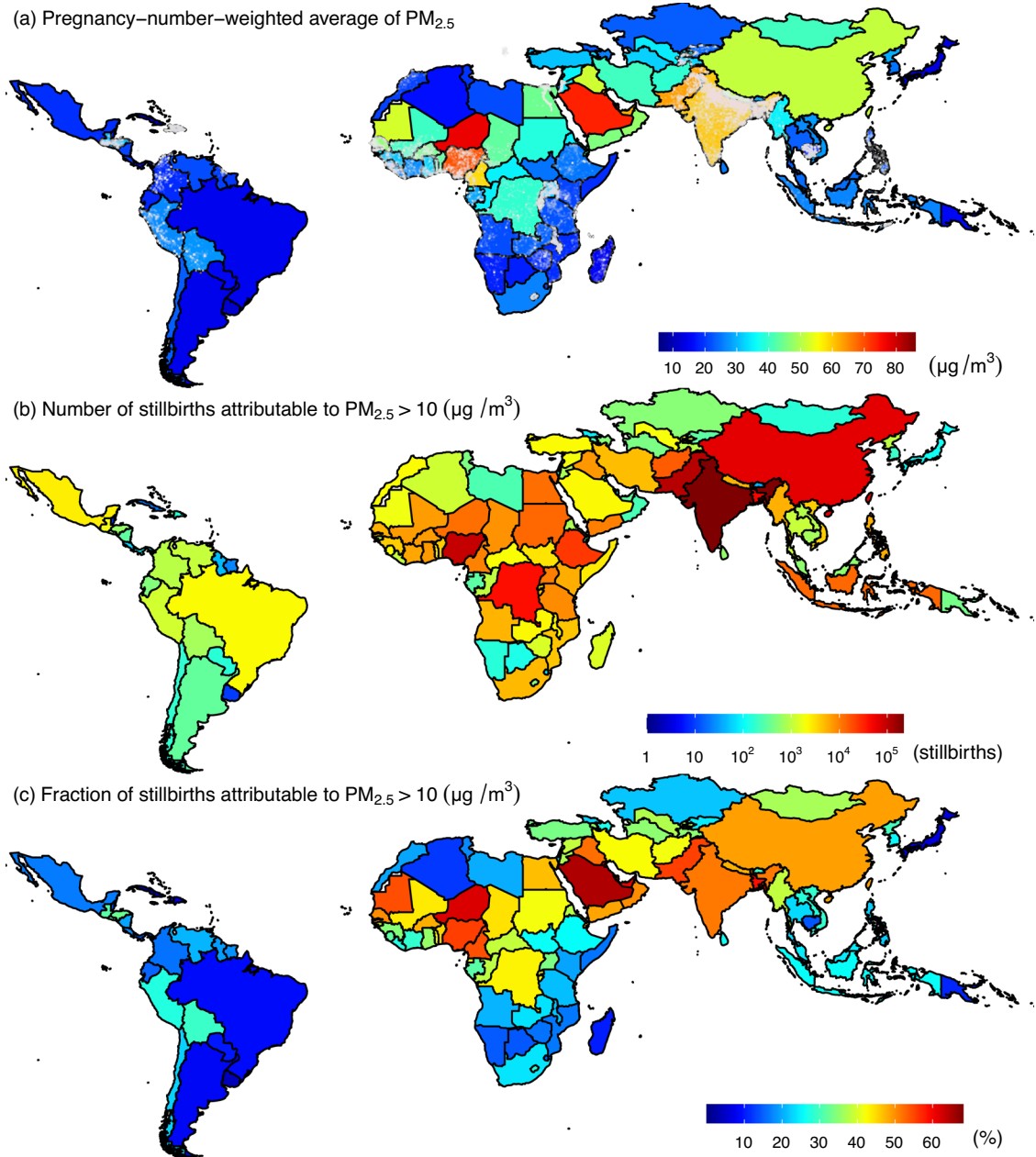

**Fig. 1 | Maps of stillbirths attributable to PM$_{2.5}$ exposure in the 137 countries.** PM$_{2.5}$ exposure (**a**), number (**b**), and fraction (**c**) of stillbirths attributable to exposure in 2015 in the 137 countries. Gray dots in panel (**a**) are the surveyed locations for the samples from 54 low- and middle-income countries used to derive the exposure–response curves.

difference could be explained by the estimated curvature of the exposure–response relationship. The sublinear curvature suggests that low-concentration exposure is more impactful than high-concentration exposure, whereas using a loose referent makes the risk assessment focus on the effect of high-concentration exposure.

Country-level estimates of the fraction and number of stillbirths attributable to PM$_{2.5}$ are shown in Fig. 1 and Supplementary Table 4. Estimates from different exposure-response curves showed a similar spatial pattern (Supplementary Figure 10). The countries with the largest numbers of PM$_{2.5}$-related stillbirths were India (217000 [95% CI: 151000, 279000]), Pakistan (110000 [95% CI: 77000, 142000]), Nigeria (93000 [95% CI: 51000, 137000]), China (64000 [95% CI: 42000, 81000]), and Bangladesh (49000 [95% CI: 35000, 61000]). The countries with the highest fraction of stillbirths attributable to PM$_{2.5}$ were Qatar (71.16% [95% CI: 56.17, 80.88]), Saudi Arabia (68.38% [95% CI: 52.37, 79.06]), Kuwait (66.08% [95% CI: 48.76, 77.87]), Niger (65.68%

[95% CI: 50.15, 76.77]), and the United Arab Emirates (64.63% [95% CI: 46.47, 76.45]). South Asia, sub-Saharan Africa, and the Arabian Desert were hotspots of PM$_{2.5}$-related stillbirths, due to high exposure and baseline stillbirth rate.

Within the study domain, the total number of stillbirths decreased at an annual rate of 1.95% (95% CI: 1.76, 2.15) from 2000 (2.83 [95% CI: 2.60, 3.08] million) to 2009 (2.37 [95% CI: 2.24, 2.49] million), and stably decreased by 2.05% (95% CI: 1.99, 2.12) from 2010 (2.31 [95% CI: 2.20, 2.43] million) to 2019 (1.93 [95% CI: 1.79, 2.05] million). By contrast, as shown in Supplementary Fig. 9, the number of stillbirths attributable to PM$_{2.5}$ (>10 µg/m$^3$) slowly decreased by 0.54% (95% CI: −0.03, 1.11) from 2000 (0.97 [95% CI: 0.59, 1.30] million) to 2009 (0.93 [95% CI: 0.61, 1.20] million), and the reduction rate increased to 2.84% (95% CI: 3.24, 2.43) from 2010 (0.90 [95% CI: 0.59, 1.15] million) to 2019 (0.71 [95% CI: 0.45, 0.92] million). These results suggest that the improved air quality in some of the 137 countries (e.g., China) might

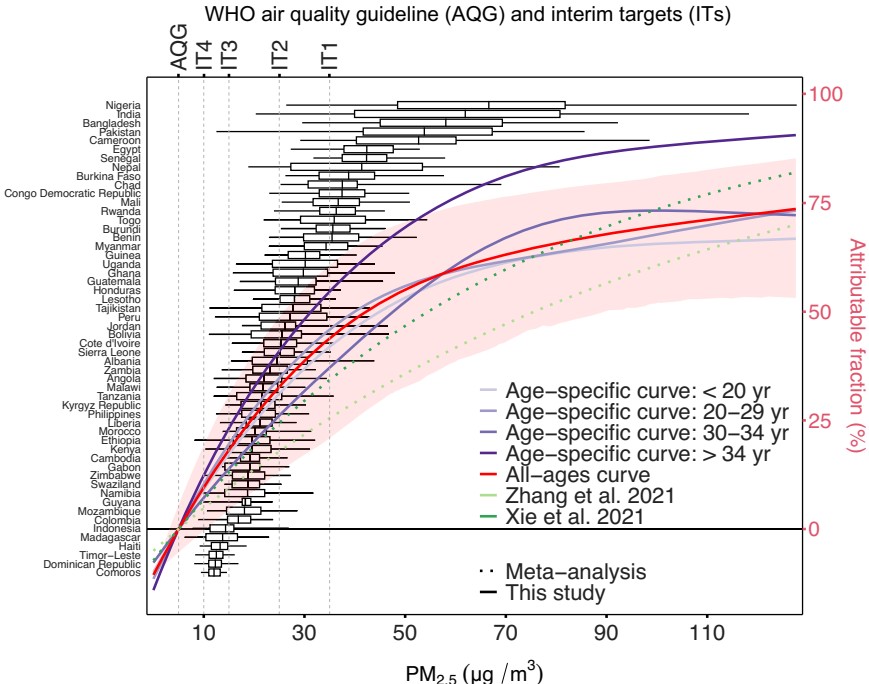

**Fig. 2 | Multiple exposure–response curves to link PM$_{2.5}$ with stillbirth (right y-axis).** The all-ages curve and age-specific curves were derived from our self-matched case-control study of stillbirths from 54 countries (left y-axis). The corresponding PM$_{2.5}$ exposures in those countries are shown as boxplots (The centre bars are medians; the box bounds and whiskers indicate for ranges from 25$^{th}$ to 75$^{th}$ and from 2.5$^{th}$ to 97.5$^{th}$ percentile, respectively.). The red ribbon is the pointwise 95% confidence interval for the all-ages curve.

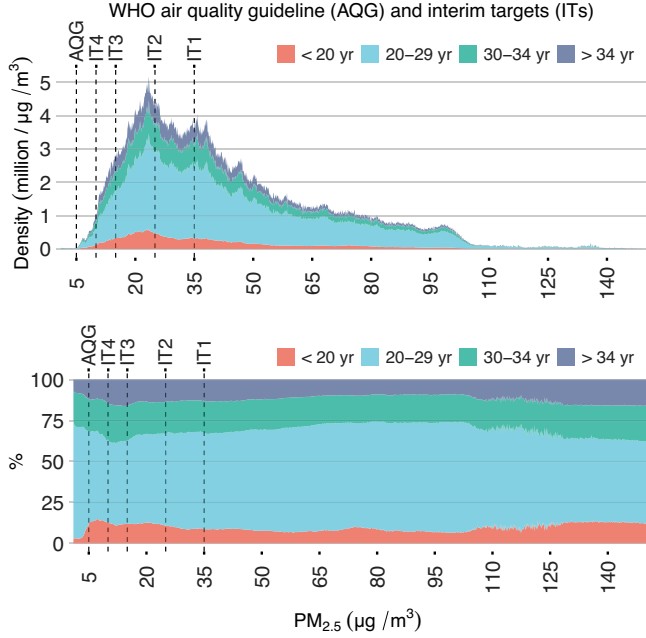

**Fig. 3 | PM$_{2.5}$ exposure distribution among the 137 countries.** Distributions (top) and relative fractions (bottom) of age-specific populations at risk (i.e., pregnancies stratified by maternal age).

underlie the reduction in the global burden of stillbirths. Therefore, meeting the WHO air quality targets could prevent stillbirths. Achieving the IT1, IT2, IT3, IT4 or AQG target for long-term PM$_{2.5}$ could reduce the number of stillbirths by 0.21 (95% CI: 0.13, 0.28) million, 0.37 (95% CI: 0.26, 0.47) million, 0.59 (95% CI: 0.39, 0.76) million, 0.71 (95% CI: 0.45, 0.92) million, or 0.83 (95% CI: 0.52, 1.08) million, respectively (Fig. 4). In terms of point-estimates, the health benefit from achieving

AQG (0.83 million reduction) is comparable to the total number of stillbirths avoided by 20-year changes in all influencing factors (0.90 million reduction).

## Discussion

We explored the association between PM$_{2.5}$ and stillbirth in LMICs and performed the first global risk assessment on PM$_{2.5}$-related stillbirths. Based on the lower boundary of the main result (i.e., referring to IT4 and using the age-specific PM$_{2.5}$–stillbirth curves), at least a quarter of stillbirths are attributable to PM$_{2.5}$ exposure during gestation.

An association between stillbirths and PM$_{2.5}$ has been reported in North America[19,20], Europe[21], East Asia[13,22,23], South Asia[14], and Africa[10]. Recently, the association was confirmed by evidence on outcomes closely related to stillbirth, including spontaneous abortion[23] and miscarriage[19]. For instance, using the same epidemiological design as this study (i.e., a case-crossover design) to analyze 3583 women in the Nurses' Health Study II, Gaskins et al.[19] reported an OR for spontaneous abortion per 2.0 μg/m$^3$ increment in PM$_{2.5}$ of 1.10 (1.04, 1.17). Due to the limited sample size, we have reported ORs for an outcome combining early stillbirth and stillbirth for each 10 μg/m$^3$ increment total PM$_{2.5}$ of 1.09 (95% CI: 1.05–1.14) in Africa[10] and 1.07 (1.02–1.12) in South Asia[14], using DHS data. In this study, which included samples from South America and Southeast Asia, and excluding early stillbirths, the OR was estimated to be 1.11 (95% CI 1.06, 1.16), consistent with previous findings.

Although potential biological mechanisms for the association between PM$_{2.5}$ exposure and pregnancy loss are not clear yet, some pathways can explain it to some extent. First, fine ambient particles may directly cross the placental barrier, and trigger hypoxic or immune-mediated injuries, which can cause irreversible embryonic damages leading to stillbirth[24]. Second, PM$_{2.5}$ exposure during pregnancy has been reported to increase the maternal methemoglobin level, which can induce fetal oxidative stresses and inhibit the oxygen transport[25]. Finally, placental abnormalities are more frequently found in stillbirths than in livebirths[26], and have been considered as possible

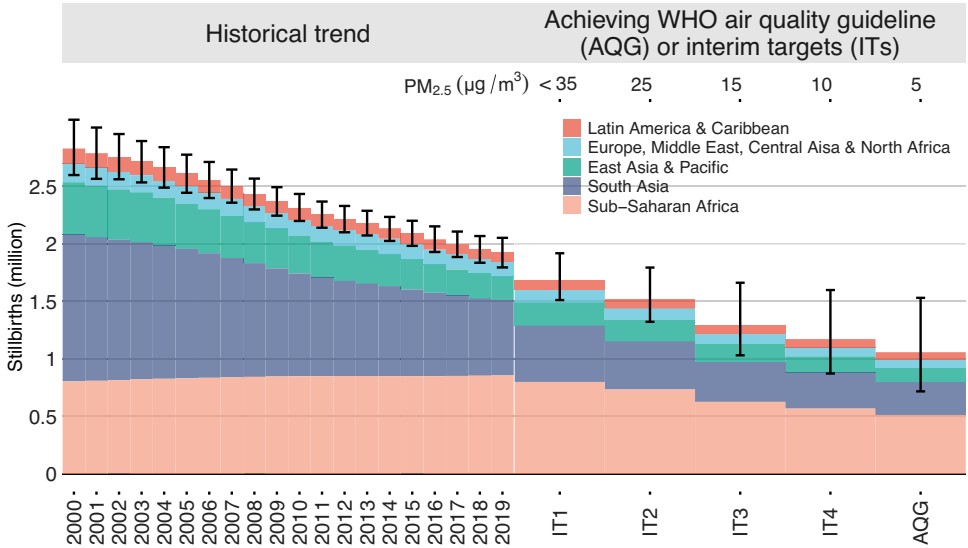

**Fig. 4 | Numbers of total stillbirths and remaining stillbirths after achieving PM$_{2.5}$ targets set by the World Health Organization (WHO) among the 137 countries from 2000 to 2019.** The effects of achieving the targets were estimated from the age-specific PM$_{2.5}$–stillbirth curves. Error bars are the corresponding 95% confidence intervals estimated by Monte Carlo approach.

mechanisms to explain the fetal deaths. PM$_{2.5}$ exposure has positively correlated with placental global DNA methylation[27], and suboptimal placental perfusion induced by the increased blood viscosity[28], both of which can lead to placental dysfunctions.

Our study adds evidence verifying the adverse effect of PM$_{2.5}$ on stillbirth, an important but neglected public health issue. According to an UN IGME study[2], although the average rate of stillbirth decreased from 2000 to 2019, the trend was slower than that of the mortality rate of children under 5 years old, an indicator of maternal health. This suggests that efforts to promote maternal health were unequal for different adverse outcomes, and interventions relevant to stillbirths are inadequate. In addition, no decrease in the stillbirth rate was found for 34 countries in sub-Saharan Africa, 16 countries in East Asia and the Pacific, and 15 countries in Latin America and the Caribbean, which reflects the geographic inequality in efforts to prevent stillbirths. Current efforts to prevent stillbirth focus on medical service improvements, such as providing ongoing intrapartum monitoring and timely intervention in cases of complications, increasing coverage of prenatal care, and strengthening emergency obstetric care and caesarean section. Furthermore, our findings show that exposure to an adverse environment during pregnancy can increase the risk of stillbirth. Compared to clinical risk factors, environmental ones are usually unseen, due to their lesser effects at the individual level. However, prolonged exposure renders them of public-health importance. Clean air policies, which have been enacted in some LMICs, such as China[29–31], can prevent stillbirths. In addition, personal protections against air pollution, i.e., wearing masks, installing air purifiers, avoiding going outside when air pollution occurs, could also protect vulnerable pregnant women.

There are some limitations of this study to mention. Those related to the development of PM$_{2.5}$–stillbirth curves, including those on the data quality (e.g., underreporting of stillbirth, misclassified outcomes and recall bias), exposure assessment (e.g., exposure misclassification introduced by residential mobility during pregnancy, and limited temporal resolution in DHS data on gestational length), and overlooked confounders (particularly, longitudinal risk factors on birth outcomes, e.g., antenatal clinic attendance[32], gestational hypertension and diabetes) are discussed in the previous reports[10,12,14] and are not repeated here. For instance, underreporting stillbirths could not be completely avoided, and the potential bias due to underreporting or misclassifying of stillbirth has been evaluated by a simulation analysis[12], which might cause an underestimated PM$_{2.5}$–stillbirth

association. Even though we used the best-available estimates on the global burden, this issue could lead to underestimated numbers of PM$_{2.5}$–related stillbirths, and the bias might be also varied between countries. The current study included all eligible controls and the unbalanced number of controls might introduce bias. However, our previous studies examined other alternative control selections (i.e., healthy control, nearby control, and nulliparous control) and found the bias had little influence on the estimated associations. Additionally, in this study, our epidemiological models had considered the heterogeneity between countries as the random effects. Some limitations as mentioned above (e.g., incomparable data quality between countries) are reasons underlying the heterogeneity. Therefore, modelling heterogeneity could help to control for potential bias. Here, we focus on the limitations of the risk assessments. First, most risk assessments, including ours, assume that exposure–response curves are generalizable. In other words, the PM$_{2.5}$–stillbirth curves derived from a sampled population were assumed to predict the risk in the general population. To improve the generalizability of the exposure–response curves used in the main assessment model, we controlled for between-country heterogeneity in the epidemiological models (Eqs. 1–2), examined whether subpopulation-specific effects of PM$_{2.5}$ were homogenous, developed age-specific assessments to address the potential heterogeneity embedded in the effect of PM$_{2.5}$, and derived sublinear curves to model the variation in the marginal effect of PM$_{2.5}$ between regions with different levels of long-term exposure. For LMICs, where stillbirths were prevalent, the above efforts made our PM$_{2.5}$–stillbirth curves more representative than those based on meta-analyses of evidence from both LMICs and high-income countries. However, the generalizability was still questionable. For instance, in most risk assessments, the toxicity of PM$_{2.5}$ is assumed to be homogenous, which may be invalid. Particularly, in the sub-Saharan Africa and Arabian Desert areas, hotspots of PM$_{2.5}$-related stillbirths (Fig. 1), PM$_{2.5}$ is rich in dust particles. Whether exposure to dust particles is associated with stillbirths is unknown and should be investigated. Therefore, risk-assessment results in those regions should be interpreted with caution. In future, data on PM$_{2.5}$ components from multiple sources should be collected and ensembled for relevant exposure and risk assessment. Second, the stillbirth risk in this study is not independent of other health impacts from PM$_{2.5}$, and thus our results should not be interpreted as an extra burden that adds to previous findings, such as preterm births attributable to PM$_{2.5}$. Short gestation,

an outcome included in GBD assessment of $PM_{2.5}$[16], increases the probability of stillbirth, and thus can act as a mediating pathway for the $PM_{2.5}$–stillbirth association. In our models, because we did not adjust for gestational duration, the exposure–response curves present the total effect of $PM_{2.5}$, including the indirect effect mediated by short gestation and the direct effect on stillbirth. Finally, due to the limited data, our assessments were annual and at the country level, limiting the policy implications of the findings. Gestational $PM_{2.5}$ exposure can vary seasonally, so that the risk of $PM_{2.5}$-related stillbirth varies throughout the year. Planning conception for a particular time can reduce gestational $PM_{2.5}$ exposure, but this requires further risk assessment studies with a fine temporal resolution. In conclusion, we developed age-specific exposure–response curves to assess the risk of stillbirth attributable to gestational $PM_{2.5}$ exposure using individual-level data from 54 LMICs and applied the curves to evaluate the disease burden in 137 countries in which 98% of global stillbirths occur. We found that $PM_{2.5}$ exposure contributed to 39.7% (95% CI: 26.1, 50.8) of stillbirths in the 137 countries. Meeting the WHO air quality targets could thus prevent a considerable number of stillbirths.

# Methods

## Study population

Valid records of pregnancy outcomes were obtained from the Demographic and Health Surveys (DHS) program from 1998 to 2016. The DHS program is an ongoing series of standardized, nationally representative surveys conducted in more than 90 countries every 5 years. In each wave, samples are obtained via a multi-stage stratified cluster sampling approach to guarantee representativeness. Women aged 15–49 years are selected for in-depth surveys on a broad range of health issues, including maternal and child health, nutrition, fertility, and reproductive health. Details of the sampling frame, survey questionnaire, and data are available after completing a simple registration process followed by a request for access on the DHS website (https://www.dhsprogram.com/). Global positioning system (GPS) devices were used to collect geospatial information for each primary sampling unit (e.g., urban wards and rural villages) in most survey waves. From 113 geocoded surveys of 54 LMICs (Fig. 1a), we extracted variables related to reproductive history and individual-level demographic features (Table 1) regarding the analyzed mothers, as described previously[10,14]. For each mother, to minimize recall bias, we incorporated only the most recent case of pregnancy loss and all available controls in the study period. Additional details on the dataset and data preparation can be found on the DHS website and in previous publications[10,14].

The major health outcome for this study was stillbirth, uniformly defined as babies born with no sign of life at ≥28 weeks of gestation for all the studied countries. Because the DHS reports gestation in months, here, all early terminated pregnancies with a gestational length ≥ 7 months were defined as stillbirths. For sensitivity analysis, we collected similar outcomes, including early stillbirth (gestational length 5–6 months), miscarriage (gestation < 5 months), and pregnancy loss (i.e., any of stillbirth, early stillbirth, or miscarriage).

Procedures and consent from all participants for DHS surveys have been reviewed and approved by ICF Institutional Review Board (IRB), and an IRB in the host country. This study is based on the publicly available DHS data, and thus no further ethic approval is required.

## Environmental variables

In previous studies, we assessed monthly exposure to $PM_{2.5}$ according to a combination of multiple datasets. Recently, a new study[33] generated state-of-the-art monthly $PM_{2.5}$ concentrations from 1998 to 2019 with a fine spatial resolution of 0.01° × 0.01° (~1 km × 1 km), by fusing chemical transport models, satellite remote sensing measurements, and geographic variables. Accordingly, we updated the exposure assessment by applying the new product, which was in good agreement with the ground-surface observations ($R^2 = 0.84$).

To control for the effects of climate on stillbirth, we obtained monthly temperatures with a spatial resolution of ~50 km from the Modern-Era Retrospective analysis for Research and Applications (MERRA-2) from 1998 to 2016. We also used a satellite nightlight as an indicator of economic growth. Nightlight is correlated with several socioeconomic factors, such as population density and the regional product value, and has been used to explore the effect of developmental level on human health[34]. We obtained annual nightlight data with a raw spatial resolution of 1 × 1 km from a harmonized product from 1998 to 2016[35].

## Epidemiological method to derive age-specific exposure–response curves

We performed a self-matched case-control study of the association between pregnancy loss and environmental exposures. The approach is documented in previous studies[10,12], and is briefly summarized here. The design is similar to (but subtly different from) a case-crossover study[36] or sibling-matched study[37], and has been used to link air pollution and adverse outcomes, including stillbirth[10,14], preterm birth[36], and low birthweight[37]. We examined how $PM_{2.5}$ exposure varied across multiple gestations of the same mother with different outcomes. Therefore, we focused on mothers who reported at least one stillbirth and created a fixed effect ($\theta$) according to the mothers' IDs. We used logit regression of the fixed effect (also known as a conditional logit regression) for epidemiological analysis. The model is as follows:

$$\text{Logit}(p_{i,t}) = x_{i,t}\,\beta + \mathbf{z}_{i,t}\,\gamma + \theta_i \qquad (1)$$

where the subscripts $i$ and $t$ denote the mother ID and time index of a specific gestation, $p_{i,t}$ denotes the probability of stillbirth, $x_{i,t}$ is the target exposure variable, $\beta$ is the coefficient of the estimated effect, $\mathbf{z}_{i,t}\,\gamma$ controls for the potential confounders, and $\theta_i$ is a nuisance parameter to control for the fixed effect. The fixed effect denotes all temporally invariant factors that affect stillbirth and includes variables such as genetics and geographic conditions (e.g., climate region). The linear effect was evaluated as the odds ratio ($OR = e^{\beta \times 10}$) of stillbirth per 10 µg/$m^3$ increase in $PM_{2.5}$. Because the spatial confounders are controlled for by the fixed effect, the covariates ($\mathbf{z}_{i,t}$) focus on longitudinal effects, including the nonlinear effect of maternal age, which is modeled as a spline with three degrees of freedom (DF), parity, the nonlinear effect of temperature as modeled by a 3-DF spline, seasonality as a 4-DF spline of month, and satellite nightlight. The covariates also included two random effect terms including the country-specific trends and country-specific slope of $PM_{2.5}$ to control for heterogeneity.

The target exposure variable ($x_{i,t}$) denotes the concentration of $PM_{2.5}$ during a hazard time-window (HTM) for stillbirths. The controls and cases were matched for the HTM length. For each mother, the HTM started from the month of conception and its duration was determined by the gestational month of stillbirth. Therefore, we used HTMs of the same length for all samples (i.e., one case and one or multiple controls) affiliated with a mother. Therefore, the exposure indicator is termed the gestation-adjusted $PM_{2.5}$, referring to a previous study[38], and we have applied it in previous analyses[12,37,39].

To estimate the age-specific nonlinear exposure–response curves, Eq. (1) was modified as follows:

$$\text{Logit}(p_{i,t}) = f_{\text{age}}(x_{i,t}) + \mathbf{z}_{i,t}\,\gamma + \theta_i \qquad (2)$$

where $f$ denotes a set of thin-plate spline functions. To develop age-specific curves, regression coefficients for the spline terms were estimated by strata of four maternal-age groups (i.e., <20 years, 20–29 years, 30–34 years, and >34 years). We also generated an exposure–response curve for all ages from an alternative model, which estimated the coefficients for spline terms without age stratification. The exposure-response functions estimated by Eq. (2)

or obtained from previous meta-analyses are documented in the Supplementary Data 1.

## Exposure and risk assessment

The population-weighted mean of $PM_{2.5}$ concentration or attributable fraction is used to present the level of exposure or relevant risk, respectively, in traditional assessment studies. For stillbirths, the population at risk was pregnant women, so the number of pregnancies, instead of the total population, should be used as the weight. Therefore, we obtained gridded maps of total pregnancy number ($P_s$) with a spatial resolution of 1 km × 1 km, in 2015, for 161 countries or regions[17]. To further quantify the age-specific populations at risk ($P_{s,k}$), in each pixel ($s$), we further divided the total number of pregnancies ($P_s$) using the following equation:

$$P_{s,k} = [(W_{s,k} \times R_{i,k})/\sum_k(W_{s,k} \times R_{i,k})] \times P_s, s \in i, \qquad (3)$$

where $s$, $k$, and $i$ denote the indexes for spatial pixel, age group, and country, respectively; $s \in i$ is pixel–country relationship (i.e., the $s^{th}$ pixel is within the $i^{th}$ country); $W_{s,k}$ is the gridded female population at a reproductive age (10–54 years); and $R_{i,k}$ is the age-specific fertility rate for all pixels in $i^{th}$ country. $W_{s,k}$ was obtained from WorldPop products on the sex/age-specific populations across the 1 km × 1 km grid in 2015[40]; $R_{i,k}$ was obtained from the estimates generated by the GBD study[41]. $P_{s,k}$ was first derived according to 5-year age groups, and then the results were aggregated into four groups (i.e., <20 years, 20–29 years, 30–34 years, and >34 years) to be matched with age-specific exposure–response curves. Finally, the age-specific populations at risk ($P_{s,k}$) were used as the weights for exposure and risk assessment. Because gridded maps of pregnancies were only available in 2015, we assumed that the weights were constant, and selected the assessments in 2015 as the main results.

The country-level exposure and risk were assessed for 2000 to 2019, using the following equations:

$$\text{Exposure}_{i,y} = \left[\sum_{s \in i}(C_{s,y} \times \sum_k P_{s,k})\right]/\left(\sum_k \sum_{s \in i} P_{s,k}\right),$$
$$\text{Exposure}_{i,y,k} = \left[\sum_{s \in i}(C_{s,y} \times P_{s,k})\right]/\left(\sum_{s \in i} P_{s,k}\right),$$
$$AF_{i,y} = \left[\sum_k \sum_{s \in i}(AF_{s,k,y} \times P_{s,k})\right]/\left(\sum_k \sum_{s \in i} P_{s,k}\right), \qquad (4)$$
$$AF_{s,k,y} = 1 - 1/\exp(f_k[\max(C_{s,y} - C_0, 0)])$$
$$AN_{i,y} = AF_{i,y} \times N_{i,y}$$

where $s$, $k$, $i$, and $y$ denote the indexes for spatial pixel, age group, country, and year, respectively; $C_{s,y}$ is the gridded annual average of $PM_{2.5}$ concentrations; $C_O$ is a referent exposure level for minimum risk; $\text{Exposure}_{i,y}$ is the average exposure level of $PM_{2.5}$ and $\text{Exposure}_{i,y,k}$ is an age-specific average; $f_k$ is an age-specific exposure–response curve, estimated from Eq. (2); $AF_{s,k,y}$ is the attributable fraction for a pixel and $AF_{i,y}$ is its country-level weighted mean; $N_{i,y}$ represents the total number of stillbirths for the $i^{th}$ country and $y^{th}$ year; and $AN_{i,y}$ is the annual and country-level number of stillbirths attributable to $PM_{2.5}$ exposure. $C_O$ was selected from the air quality guideline (AQG) or interim targets (IT) on long-term concentration of $PM_{2.5}$, launched by the World Health Organization (WHO) in 2021[42]. Due to the computational burden, the empirical confidence intervals (CI) for $AF_{i,y}$ and $AN_{i,y}$ only considered the uncertainties embedded in the exposure–response curves and stillbirth baseline ($N_{i,y}$). We used a Monte Carlo approach to simulate the corresponding CIs. In sensitivity analyses, we replaced the age-specific exposure–response curves by (1) the all-ages curve estimated in this study, (2) the log-linear curve from the meta-analysis by Zhang et al.[15] or (3) a log-linear curve from Xie et al.[11].

All analyses were run in R software. Statistical inference for the regression models was performed using the R package *survival*.

## Reporting summary

Further information on research design is available in the Nature Research Reporting Summary linked to this article.

## Data availability

The datasets used in this study are publicly available. The Demographic and Health Survey (DHS) dataset, $PM_{2.5}$ dataset, MERRA-2, stillbirth number, and population datasets of pregnancies and age-specific females are available from https://www.dhsprogram.com/, https://sites.wustl.edu/acag/datasets/surface-pm2-5/, https://disc.gsfc.nasa.gov/datasets?project=MERRA-2, http://childmortality.org/, and https://www.worldpop.org/, respectively.

## Code availability

The R codes for the epidemiological analyses are documented in the Supplementary Data 2. The R codes and relevant data to reproduce the figures are also within Supplementary Data 2.

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

## Acknowledgements

This work was supported by National Natural Science Foundation of China (42175182, T.X.), PKU-Baidu Fund (2020BD031, T.X.), Energy Foundation (G-2208-34045, G-2107-33169, and R-2109-33379, T.Z.), and CAMS Innovation Fund for Medical Sciences (2017-I2M-1-004, T.G.).

## Author contributions

Study design: T.X.; Draft of the manuscript: T.X. & M.T.; Data preparation: T.X., Jia.L., He.L., R.W., Ho.L. & Y.L.; Data analysis: T.X. & M.T.; Result interpretation: T.G., Jiw.L., & P.L.; Conception of the study: T.X. & T.Z.; All co-authors revised the manuscript together.

## Competing interests

The authors declare no competing interests.
