## [Peer Review File · Nature Communications]

Estimation of Stillbirths Attributable to Ambient Fine Particles in 137 CountriesREVIEWER COMMENTS

Reviewer #1 (Remarks to the Author):

Xue et al. present a multi-country risk assessment of PM2.5-associated stillbirths, using an exposure-response function derived by themselves. They found ~ 40% of stillbirths were attributable to PM2.5. The relationships between PM2.5 and stillbirth are an interesting topic and this study comes in time. This study is advantage of its large sample size, advanced epidemiological design and a batch of sensitivities analyses. However, the methods and datasets utilized in this study are complex. Some technical details are not clear and should be added to the manuscript, before publication.

Major comments:

1. The authors have published relevant studies in Africa and India. This study included the datasets analyzed by the authors before. I understand that this study present new results. However, the authors should give more details to help the readers to distinguish the differences between the current study and their previous ones.
2. Based on the Figure S1, it seems their association analyses are sensitive to adjustment for heterogeneity. Therefore, the heterogeneity in effect of PM2.5 should be presented. I suggest the authors to show some results on country-level associations. What's are the explanations for the heterogeneity?
3. It is interesting that the authors present the estimated burden of stillbirths using different exposure-response relationships, including the those derived by themselves and those by the meta-analyses. The authors only compared the total estimates from different relationships. They should present more results on that. Would different relationships change the spatial pattern of PM2.5-related stillbirths?
4. The recall bias could differ between stillbirth and livebirths because stillbirth was restricted to be the most recent case but all control livebirths were considered. Was there any concern about this?
5. For retrospectively sampled pregnancies, the average time interval between livebirth and stillbirth analyzed in this study should be reported, and was there any concern about change in residence?
6. The authors should consider more confounders in the manuscript, such as disease history (such as diabetes , pre-pregnancy BMI (such as obese status), nutritional factors, lifestyles (such passive or active smoking, drinking status pre or during pregnancy), and mental health which might be risk factors of stillbirth.

Minor comments:

1. The abstract should state how many mothers or how many case and control pregnancies were used in this study.
2. The environmental variables are not fully described in the text, please add the more detailed information, e.g., mean/max/minimum temperature over the pregnancy?
3. The authors should document their derived exposure-response functions as a look-up table, as those presented in GBD studies.
4. Line 165, The equation (1) should be revised. i and t should be shown as footnotes.
5. Line 242-243 The units for nightlight data seem to be incorrect. Similar problem in Table 1.
6. Please add the unit "years" of categorical variables of maternal age in Table 1.
7. The column names in Table 1 might cause misunderstanding, "stillbirth" column means all case and control pregnancies analyzed about the stillbirth outcome.
8. Line 267 The caption of Figure 1 seem to be incorrect and should be "number (b), and fraction (c)".
9. The legend in Figure 2 should be consistent with Figure 3 and Figure 4, which should be "WHO air quality guideline (AQG) and interim targets (ITs)"
10. Line 303 the word "he" is a typo and should be corrected.
11. It would be helpful to provide a more through discussion of the mechanisms for the association between PM2.5 and pregnancy loss.
12. Please add the specific definition of low-, middle-, and high-nightlight groups in Figure S2.
13. In Line 240, standard deviation [SD], should be revised as (SD). Please put SD in bracket.
14. Lines 241-247, authors made a comparison in terms of Nightlight and PM2.5 levels between the case and control groups. However, there was no information on whether the authors made the comparison test? Please add the P-value of the comparison test in Table 1.
15. Figure 1 presents "PM2.5 exposure (a), fraction (b), and number (c) of stillbirths attributable to

exposure in 2015 in the 137 studied countries”, but in Lines 247-248, the description of “54 LMICs” seem inconsistent with 137 countries? Please clarify.

16. Figure 4, “...from 2001 to 2019” should be “...from 2000 to 2019”

17. Line 213-224 and Table S3, is the description of “county level” an error? Should It be “country level”?

18. I suggest the authors add a flow chart of the study subjects analyzed in this study if possible because there occur many different time intervals such as 1998-2016, 2000-2019, 2003-2014, which makes the readers confused about whether the sample size changed for the different time interval.

19. Conclusions section: The authors concluded that “We found that PM2.5 exposure contributed to 39.7% (95% CI: 26.1, 50.8) of stillbirths per year in the 137 countries”, whereby the description of “per year” might be incorrect in that the authors only analyzed the year of 2015.

20. Table S2 and Table S3: I suggest the authors add the full name of the abbreviation such as ISO, AFG, AGO... in the footnote of Table S3, and the column name of the table should be continued on each page for the table S3 is too large. Please try to make the presentations of each table and figure easily understand.

21. Figure S3: the resolution should be improved and the Y-axis is not so clear.

Reviewer #2 (Remarks to the Author):

Using DHS data, Xue and colleagues examined the association between exposure to ambient fine particulate matter and the risk of stillbirths. They used the estimated dose-response curves to estimate the burden of stillbirths attributable to ambient fine particulate issues in 137 countries. The manuscript was generally well-written. However, I have a few major concerns:

1. Assuming the dose-response relationship between exposure to fine particulate matter and stillbirths estimated from DHS countries is the same as that in 137 countries is controversial and might bias the burden of stillbirths attributable to fine particulate matter. First, the components of PM2.5 in DHS countries are different from other developed countries. Dust accounts for a large proportion of PM2.5 in DHS and other countries are mainly traffic-related. The toxicology of PM2.5 varied by location, especially between DHS countries and developed countries. In the manuscript, the authors did not consider this an important issue.

2. The authors selected all eligible controls and compared the concentration of PM2.5 between controls and cases. The number of controls ranges from 1 to 5+. This unbalanced number of controls might introduce bias.

3. The underreporting of stillbirths is a key limitation of the data.

Reviewer #3 (Remarks to the Author):

The manuscript presents an interesting analysis attempting to quantify the burden of stillbirths directly attributed to air pollution (ambient fine particles) across 137 Countries using DHS and other estimates of exposure for pollution and other fixed effects. The builds on the back of recent studies which have demonstrated an epidemiological association between gestational exposure to fine particulate matter (PM2.5) and stillbirths. The framework used has been developed and tested previously as well as published. This work is an important contribution to this area and provides an usual estimation of the attributable stillbirth burden at high resolution across LMIC. have some comments related to the inherent biases related to stillbirth ascertainment using these type of data as well as some further comments below.

Outcomes such as stillbirths recorded/reported through DHS have know biases related vital status (and missed events), inaccuracies related to gestational age estimation etc. How did you adjust or account for these prior to your associative modelling step? Similarly, were any adjustments made for locations/contexts where home births may have been more prevalent and potentially impact this may have had on the stillbirth outcome ascertainment?

How did you account for potentially alternative stillbirth definitions and source types across the various DHS?

Was reported antenatal clinic attendance incorporated or tested in your model? E.g.

<https://bmjopen.bmj.com/content/12/2/e051675>

This would be an important confounder to account for when testing the primary hypothesis.

Fixed effect – genetics (line 170) – unclear what this means and how this was measured/quantified?

As is now standard practice re: GATHER compliance – I would suggest including a completed GATHER statement as part of the supplementary materials and also providing a link to a GitHub repository (or similar platform) with all the data processing and modelling scripts.

Dear reviewers,

Thank you for giving us the opportunity to submit a revised draft of the manuscript "Estimation of Stillbirths Attributable to Ambient Fine Particles in 137 Countries". We really appreciate the time and effort that you dedicated to improving our paper and are grateful for the valuable comments and suggestions. We have read all comments and suggestions deliberately and incorporated most of the suggestions. Those changes are tracked in the manuscript, and the detailed point-by-point responses to the reviewers' concerns and comments are listed as follows, and the direct corrections are highlighted by blue color. For easy reading, we also submit a clean version of manuscript, and the reference citations and line numbers shown in following point-by-point responses are consistent with this clean version.

Reviewer #1 (Remarks to the Author):

Xue et al. present a multi-country risk assessment of PM_{2.5}-associated stillbirths, using an exposure-response function derived by themselves. They found ~ 40% of stillbirths were attributable to PM_{2.5}. The relationships between PM_{2.5} and stillbirth are an interesting topic and this study comes in time. This study is advantage of its large sample size, advanced epidemiological design and a batch of sensitivities analyses. However, the methods and datasets utilized in this study are complex. Some technical details are not clear and should be added to the manuscript, before publication.

Major comments:

1. The authors have published relevant studies in Africa and India. This study included the datasets analyzed by the authors before. I understand that this study present new results. However, the authors should give more details to help the readers to distinguish the differences between the current study and their previous ones.

[Response] Suggestion accepted. We added a few sentences and explained the statement clearly in the introduction. Now, the paragraph read as "We have developed a self-matched case-control method to evaluate the association between PM_{2.5} and stillbirths in Africa and South Asia. The method has been recommended as a cost-effective approach to developing an exposure-response curve for PM_{2.5} and stillbirth from large-population data in LMICs. **This study aims to present an assessment study to quantify the burden of PM_{2.5}-related stillbirth, using exposure-response curves derived from our approach or other meta-analyses. Although establishing exposure-response curves was not within our major study aim, this study increased confidence on estimates by enlarging sample size, and specified the curves by age groups, compared to our previous analyses.**" (line 74 - 78).

2. Based on the Figure S1, it seems their association analyses are sensitive to adjustment for heterogeneity. Therefore, the heterogeneity in effect of PM_{2.5} should be presented. I suggest the authors to show some results on country-level associations. What's are the explanations for the heterogeneity?

[Response] Suggestion accepted. First, we added a figure to present the country-specific estimates on the association between PM_{2.5} and stillbirth. For convenience of the reviewers, the figure is also shown here.

Figure S1 The linear association between PM_{2.5} and stillbirth, estimated by countries.

Second, we believe some of the heterogeneity can be explained by measured factors, including effect modifiers, and nonlinear curvature. The heterogeneity can also be caused by unmeasured factors, including PM_{2.5} chemical species and different data quality between DHS surveys. To mention that, we add a few words, read as “Generally speaking, the nonlinear models showed a sublinear curvature of the exposure–response relationships for both stillbirth and the secondary outcomes (Supplementary Fig. 4–Fig. 5). At high exposure levels, the modification by age tended to be apparent (Figures 2 and Supplementary Fig. 5). Therefore, a combination of the nonlinear curvature and the age’s modification could partially explain the heterogeneity between the country-specific linear associations between PM_{2.5} and stillbirth (Supplementary Fig. 6).” (line 138-140), and “Additionally, in this study, our epidemiological models had considered. Some limitations as mentioned above (e.g., incomparable data quality between countries) are reasons underlying the heterogeneity.” (line 286-289). Third, compared to the secondary outcomes (e.g., early stillbirth), our primary result on stillbirth was the most robust, given the adjustment for heterogeneity or not. The results give us some confidences that our findings may not be dramatically influenced by the heterogeneity.

3. It is interesting that the authors present the estimated burden of stillbirths using different exposure-response relationships, including the those derived by themselves and those by the meta-analyses. The authors only compared the total estimates from different relationships. They should present more results on that. Would different relationships change the spatial pattern of PM_{2.5}-related stillbirths?

[Response] Suggestion accepted. We present the spatial pattern estimates from different exposure-response curves in a supplemental figure. For reading convenience, the figure is shown as follows. To describe the results, we add a few words, which read as “Country-level estimates of the fraction and number of stillbirths attributable to PM_{2.5} are shown in Figure 1 and Supplementary Table 4. Estimates from different exposure-response curves showed a similar spatial pattern (Supplementary Fig. 10).” (line 200-201).

Figure S10 The spatial distribution of PM_{2.5}-related stillbirths, estimated by different exposure-response curves.

4. The recall bias could differ between stillbirth and livebirths because stillbirth was restricted to be the most recent case but all control livebirths were considered. Was there any concern about this?

[Response] Thank you for the comment. We agree with the reviewer that recall bias might be different between cases and healthy controls based on the self-matched case-control study design, but we believe that the bias has little effect based on our previous evidences in Africa and South Asia. In the sensitivity analyses of these two studies, we re-estimated and reported the results of different selection methods of controls, including (a) healthy controls (defined as the successful delivery of an infant who survived more than 1 year); (b) nearby controls (defined as the successful delivery occurring closest to the time of stillbirth); (c) nulliparous control (defined as the first normal delivery of the mother). The recall bias might be minimum between cases and nearby controls. Similar to the main results, we found the estimated association was not sensitive to alternative methods of control selection. Since the current study aims to estimate the global burden of stillbirth, we didn't repeat those sensitivity analyses that focusing on the associations.

5. For retrospectively sampled pregnancies, the average time interval between livebirth and stillbirth analyzed in this study should be reported, and was there any concern about change in residence?

[Response] Suggestion accepted. We documented the statistics on time intervals in Supplementary Table 1. To mention that, we added a new sentence, which reads as “The mean length of intervals between stillbirth and livebirth was 3.81 (SD = 2.45) years (Supplementary Table 2).” (line 88-89). We agree that the ignorance of migration of mother during their gestation or during study period could result in exposure misclassification. To clearly address this problem, we added a few words in the limitation. Now, it read as “Those related to the development of PM_{2.5}-stillbirth curves, including those on the data quality (e.g., underreporting of stillbirth, misclassified outcomes and recall bias), exposure assessment (e.g., exposure misclassification introduced by residential mobility during pregnancy, and limited temporal resolution in DHS data on gestational length), and study design (e.g., overlooked confounders) are discussed in previous reports and are not repeated here.” (line 280-281).

Additionally, if the change in residence affected our findings, the estimates might be considerably varied between regions with different developmental levels. However, we found the estimates were consistent between urban and rural areas or between regions with different levels of nightlight (a satellite indicator for development), as shown in Supplementary Fig.3. The homogeneity in our estimates add our confidence on that change in residence is not a problem in this study.

6. The authors should consider more confounders in the manuscript, such as disease history (such as diabetes, pre-pregnancy BMI (such as obese status), nutritional factors, lifestyles (such as passive or active smoking, drinking status pre or during pregnancy), and mental health which might be risk factors of stillbirth.

[Response] Thank you for the comment.

First, we agree with the reviewer that omitting the longitudinal covariates is a key limitation in our study. However, the longitudinal variables are not available in DHS surveys. DHS data were collected in a cross-sectional way and each mother was visited only once. The reproductive history questionnaire collected the longitudinal variables focusing on birth outcomes, such as livebirth or stillbirth, rather than their risk factors. To clarify that, in the limitation section, we add a few words, which read as “Those related to the development of PM_{2.5}–stillbirth curves, including those on the data quality (e.g., underreporting of stillbirth, misclassified outcomes and recall bias), exposure assessment (e.g., exposure misclassification introduced by residential mobility during pregnancy, and limited temporal resolution in DHS data on gestational length), and overlooked confounders (particularly, longitudinal risk factors on birth outcomes, e.g., antenatal clinic attendance, gestational hypertension and diabetes) are discussed in previous reports and are not repeated here.” (line 282-284).

Second, one aim of developing the self-matched design is to overcome the weaknesses brought by the omitted longitudinal factors. For instance, some lifestyles may not be varied dramatically within the same mother, and thus can be controlled for by the design itself. Third, we don’t worry about the health-related longitudinal factors, such as the weight gain during pregnancy, gestational hypertension, gestational diabetes, mental health during pregnancy. If the health disorders that occurred before births are related to PM_{2.5} exposure, they can act as mediators rather than confounders in our study. Therefore, we actually estimated the total effect of PM_{2.5} on stillbirth. If the health-related factors are not related to PM_{2.5} exposure, they are not confounders.

Minor comments:

1. The abstract should state how many mothers or how many case and control pregnancies were used in this study.

[Response] Suggestion accepted. We added the number of cases and controls in the abstract and it reads as “13,870 stillbirths and 32,449 livebirths were extracted from 113 geocoded surveys from the Demographic and Health Surveys. Each stillbirth was compared to livebirth(s) of the same mother using a conditional logit regression.” (line 27-29).

2. The environmental variables are not fully described in the text, please add the more detailed information, e.g., mean/max/minimum temperature over the pregnancy?

[Response] Suggestion accepted. We add detailed information on the environmental variables and present those statistics by boxplots in Supplementary Fig. 1. To mention that, we add a few words, which read as “Country-specific distributions for the environmental variables are shown in Supplementary Fig. 1.” (line 97-98). For your convenience, the figure is also documented here. The figure shows the mean and range of temperature over the pregnancy, as the reviewer suggested.

3. The authors should document their derived exposure-response functions as a look-up table, as those presented in GBD studies.

[Response] Suggestion accepted. We submitted the exposure-response functions as a series of look-up tables for the attributable fractions by PM_{2.5} levels, methods and simulations. To mention that, we add a few words, which read as “The exposure-response functions estimated by equation (2) or obtained from previous meta-analyses are documented in the supplementary data.” (line 414-415).

4. Line 165, The equation (1) should be revised. i and t should be shown as footnotes.

[Response] Suggestion accepted. We have revised the equation.

5. Line 242-243 The units for nightlight data seem to be incorrect. Similar problem in Table 1.

[Response] Suggestion accepted. We have revised the units of nightlight and it read as “Nightlight, as an indicator of developmental level, was 8.29 digit-number (DN) (SD = 16.14 DN) for the case group, higher than that for the control group (mean = 7.07 DN; SD = 14.99 DN).” (line 92-93).

6. Please add the unit “years” of categorical variables of maternal age in Table 1.

[Response] Suggestion accepted. We have revised the table.

7. The column names in Table 1 might cause misunderstanding, “stillbirth” column means all case and control pregnancies analyzed about the stillbirth outcome.

[Response] Suggestion accepted. We have added a label for the title of columns as “Outcomes and their matched controls” in the table 1.

8. Line 267 The caption of Figure 1 seem to be incorrect and should be “number (b), and fraction (c)”.

[Response] Suggestion accepted. We have revised the caption of Figure 1.

9. The legend in Figure 2 should be consistent with Figure 3 and Figure 4, which should be “WHO air quality guideline (AQG) and interim targets (ITs)”

[Response] Suggestion accepted. We have revised the legend of Figure 2.

10. Line 303 the word “he” is a typo and should be corrected.

[Response] Suggestion accepted. We have revised the word as “the”.

11. It would be helpful to provide a more through discussion of the mechanisms for the association between PM_{2.5} and pregnancy loss.

[Response] Suggestion accepted. we add a paragraph in the discussion section, which read as

“Although potential biological mechanisms for the association between PM_{2.5} exposure and pregnancy loss are not clear yet, some pathways can explain it to some extent. First, fine ambient particles may directly cross the placental barrier, and trigger hypoxic or immune-mediated injuries, which can cause irreversible embryonic damages leading to stillbirth.²⁴ Second, PM_{2.5} exposure during pregnancy has been reported to increase the maternal methemoglobin level, which can induce fetal oxidative stresses and inhibit the oxygen transport.²⁵ Furthermore, placental abnormalities are more frequently found in stillbirths than in livebirths,²⁶ and have been considered as possible mechanisms to explain the fetal deaths. PM_{2.5} exposure has positively correlated with placental global DNA methylation²⁷, and suboptimal placental perfusion induced by the increased blood viscosity²⁸, both of which can lead to placental dysfunctions.” (line 250-260).

12. Please add the specific definition of low-, middle-, and high-nightlight groups in Figure S2.

[Response] Suggestion accepted. We added the caption of that Figure, and it read as “Figure S3 The subpopulation-specific linear associations between PM_{2.5} exposure and stillbirth or secondary outcomes, including early stillbirth, miscarriage and pregnancy loss. The nightlight was classified as low (≤ 4 DN), middle (4 – 20.5 DN), or high (> 20.5 DN) level group.”.

13. In Line 240, standard deviation [SD], should be revised as (SD). Please put SD in bracket.

[Response] Suggestion accepted.

14. Lines 241-247, authors made a comparison in terms of Nightlight and PM2.5 levels between the case and control groups. However, there was no information on whether the authors made the comparison test? Please add the P-value of the comparison test in Table 1.

[Response] Suggestion accepted. The relevant sentences now read as “Nightlight, as an indicator of developmental level, was 8.29 digit-number (DN) (SD = 16.14 DN) for the case group, higher than that for the control group (mean = 7.07 DN; SD = 14.99 DN; p -value $< 2 \times 10^{-16}$ for a paired test). This suggests that more stillbirths occur in more developed regions. The controls had a lower level of gestational exposure to PM_{2.5} (mean = 40.34 $\mu\text{g}/\text{m}^3$, SD = 22.20 $\mu\text{g}/\text{m}^3$), compared to stillbirth cases (mean = 40.96 $\mu\text{g}/\text{m}^3$; SD = 23.04 $\mu\text{g}/\text{m}^3$; p -value $< 2 \times 10^{-16}$ for a paired test).”

15. Figure 1 presents “PM2.5 exposure (a), fraction (b), and number (c) of stillbirths attributable to exposure in 2015 in the 137 studied countries”, but in Lines 247-248, the description of “54 LMICs” seem inconsistent with 137 countries? Please clarify.

[Response] Suggestion accepted. We have revised that the 54 LMICs was showed on Figure 1(a). It read as “The spatial distributions of the surveyed samples from the 54 LMICs are shown in Figure 1(a), along with the exposure level of PM_{2.5} in 2015.”. Additionally, the caption of Figure 1 was revised as “Figure 1. PM_{2.5} exposure (a), number (b), and fraction (c) of stillbirths attributable to exposure in 2015 in the 137 countries. Gray dots in panel (a) are the surveyed locations for the samples from 54 low- and middle-income countries used to derive the exposure–response curves.” (line 122).

16. Figure 4, “...from 2001 to 2019” should be “...from 2000 to 2019”

[Response] Suggestion accepted. We have revised it.

17. Line 213-224 and Table S3, is the description of “county level” an error? Should It be “country level”?

[Response] Suggestion accepted. We have revised it.

18. I suggest the authors add a flow chart of the study subjects analyzed in this study if possible because there occur many different time intervals such as 1998-2016, 2000-2019, 2003-2014, which makes the readers confused about whether the sample size changed for the different time interval.

[Response] We apologized for the typo. 2003-2014 and 2000-2019 were typos, and the time intervals should be 1998-2016. In our previous studies, we utilized a dataset of PM_{2.5} since 2000. In this study, as mentioned, we used a new dataset since 1998. We collected DHS data until 2016. Therefore, the study period was from 1998 to 2016.

19. Conclusions section: The authors concluded that “We found that PM_{2.5} exposure contributed to 39.7% (95% CI: 26.1, 50.8) of stillbirths per year in the 137 countries”, whereby the description of “per year” might be incorrect in that the authors only analyzed the year of 2015.

[Response] Suggestion accepted. We removed the words, “per year”.

20. Table S2 and Table S3: I suggest the authors add the full name of the abbreviation such as ISO, AFG, AGO... in the footnote of Table S3, and the column name of the table should be continued on each page for the table S3 is too large. Please try to make the presentations of each table and figure easily understand.

[Response] Suggestion accepted. We added the ISO codes for the 137 countries as a new table (Supplementary Table 1). We also revised the column names as suggested.

21. Figure S3: the resolution should be improved and the Y-axis is not so clear.

[Response] Suggestion accepted. We modified the figure as suggested.

Reviewer #2 (Remarks to the Author):

Using DHS data, Xue and colleagues examined the association between exposure to ambient fine particulate matter and the risk of stillbirths. They used the estimated dose-response curves to estimate the burden of stillbirths attributable to ambient fine particulate issues in 137 countries. The manuscript was generally well-written. However, I have a few major concerns:

1. Assuming the dose-response relationship between exposure to fine particulate matter and stillbirths estimated from DHS countries is the same as that in 137 countries is controversial and might bias the burden of stillbirths attributable to fine particulate matter. First, the components of PM_{2.5} in DHS countries are different from other developed countries. Dust accounts for a large proportion of PM_{2.5} in DHS and other countries are mainly traffic-related. The toxicology of PM_{2.5} varied by location, especially between DHS countries and developed countries. In the manuscript, the authors did not consider this an important issue.

[Response] We agree with the reviewer that the generalizability of the exposure-response curve is questionable due to the limited knowledge on toxicology of PM_{2.5}. As mentioned by the reviewer, dust is a good example and may be more toxic than the PM_{2.5} produced by anthropogenic sources. Actually, even for the dust particles, their effects are varied by locations, because their chemical species are determined by crustal elements, and the transported dust particles can also bring complex chemicals along their traveling pathways. Therefore, the generalizability of PM_{2.5} toxicity is always questionable.

However, in purpose of regularization or taking-action, the total concentration of PM_{2.5} has been utilized a metric to evaluate the corresponding health impacts. For instance, the WHO air quality guidelines recommend a level on total concentration of PM_{2.5}, and conclude that achieving it can avoid 80% of deaths related to PM_{2.5} (<https://www.who.int/news/item/22-09-2021-new-who-global-air-quality-guidelines-aim-to-save-millions-of-lives-from-air-pollution>). Additionally, the global burden of diseases studies evaluated the health impacts of air pollution using concentrations of PM_{2.5} and O₃ as exposure metrics ([https://doi.org/10.1016/S0140-6736\(20\)30752-2](https://doi.org/10.1016/S0140-6736(20)30752-2)). Furthermore, the lack of environmental data on PM_{2.5} components (because of the high cost to monitor PM_{2.5} species routinely) also limit the risk assessment fully considering the heterogenous toxicity. Therefore, in most of extent studies, using the total concentration of PM_{2.5} to present the average health effect is a pragmatic approach to deliver epidemiological findings to the public and policymakers.

To clarify the weakness, in the limitation section, we add a few words, which read as “in most risk assessments, the toxicity of PM_{2.5} is assumed to be homogenous, which may be invalid. Particularly, in the sub-Saharan Africa and Arabian Desert areas, hotspots of PM_{2.5}-related stillbirths (Figure 1), PM_{2.5} is rich in dust particles. Whether exposure to dust particles is associated with stillbirths is unknown and should be investigated. Therefore, risk-assessment results in those regions should be interpreted with caution. In future, data on PM_{2.5} components from multiple sources should be collected and ensembled for relevant exposure and risk assessment.” (line 309-310).

2. The authors selected all eligible controls and compared the concentration of PM_{2.5} between controls and cases. The number of controls ranges from 1 to 5+. This unbalanced number of controls might introduce bias.

[Response] Thank for the comment. We agree with the reviewer that unbalanced number of controls might introduce bias. Actually, we detailed examined the bias in our previous epidemiological studies in Africa and South Asia. We found the bias had little influence on the estimated associations. Specifically, apart from choosing all available livebirth(s), in the sensitivity analyses of the two previous studies, we re-estimated and reported the results based on different selections of controls, including (a) healthy controls (defined as the successful delivery of an infant who survived more than 1 year); (b) nearby controls (defined as the successful delivery occurring closest to the time of stillbirth); (c) nulliparous control (defined as the first normal delivery of the mother). Similar to the main results, we found the estimated association was not sensitive to alternative methods of control selection. However, this study was focused on risk assessment, not the re-analysis of a dataset

combining those utilized in our previous works. For concise purpose, we didn't repeat those analyses in this paper. We would like to make further modification or explanation, if the reviewer has more considerations on this issue.

3. The underreporting of stillbirths is a key limitation of the data.

[Response] Thank for the comment. We agree with the reviewer and declare this limitation. To clarity that, we add a few words, which read as “Those related to the development of PM_{2.5}–stillbirth curves, including those on the data quality (e.g., underreporting of stillbirth, misclassified outcomes and recall bias), exposure assessment (e.g., exposure misclassification introduced by residential mobility during pregnancy, and limited temporal resolution in DHS data on gestational length), and overlooked confounders (particularly, longitudinal risk factors on birth outcomes, e.g., gestational hypertension and diabetes) are discussed in previous reports^{10,12,14} and are not repeated here. For instance, the potential bias due to underreporting or misclassifying of stillbirth has been evaluated by a simulation analysis¹², and might cause an underestimated PM_{2.5}–stillbirth association.” (line 286-288).

The simulation analysis (Supplementary Fig. 2 in our South Asia study, Xue et al. *Lancet Planet Health* 2021; **5**(1): e15-e24.) shows that underreporting of stillbirth didn't result in a biased estimate on effect of PM_{2.5}, but an underestimated burden of diseases, when using the DHS data as the input for total number of stillbirth (e.g., baseline level of risk). To avoid the underestimation, this study used the recently-developed UN IGME product as the baseline risk, which made it different from our previous analysis in the South Asia study.

Reviewer #3 (Remarks to the Author):

The manuscript presents an interesting analysis attempting to quantify the burden of stillbirths directly attributed to air pollution (ambient fine particles) across 137 Countries using DHS and other estimates of exposure for pollution and other fixed effects. The builds on the back of recent studies which have demonstrated an epidemiological association between gestational exposure to fine particulate matter (PM_{2.5}) and stillbirths. The framework used has been developed and tested previously as well as published. This work is an important contribution to this area and provides an usual estimation of the attributable stillbirth burden at high resolution across LMIC. have some comments related to the inherent biases related to stillbirth ascertainment using these type of data as well as some further comments below.

1. Outcomes such as stillbirths recorded/reported through DHS have know biases related vital status (and missed events), inaccuracies related to gestational age estimation etc. How did you adjust or account for these prior to your associative modelling step? Similarly, were any adjustments made for locations/contexts where home births may have been more prevalent and potentially impact this may have had on the stillbirth outcome ascertainment?

[Response] Suggestion accepted. We enhance the relevant discussions in the modified paper.

We agree with the reviewer that the data quality issues in DHS's records on stillbirth can lead to a misclassification outcome, which may cause a biased estimator in the associative step. According to the epidemiological theory (Rothman, Modern Epidemiology, 3rd edition), misclassification of outcomes can introduce bias into a study, but it usually has much less of an impact than misclassification of exposure. Usually, the misclassification of exposure doesn't affect the estimated association or underestimated it. In our previous study (Xue et al. *Lancet Planet Health* 2021; 5(1): e15-e24.), we further explored the potential problems in the stillbirth data using a simulation method. The potential issues include (1) underreporting of stillbirth and (2) misclassifying stillbirth with some similar outcomes (e.g., miscarriage). We found the former problem had no influence on the estimated association, and the latter might result in an underestimated association.

To clarity that, in the limitation section, we add a few words, which read as "those related to the development of PM_{2.5}-stillbirth curves, including those on the data quality (e.g., underreporting of stillbirth, misclassified outcomes and recall bias), exposure assessment (e.g., exposure misclassification introduced by residential mobility during pregnancy, and limited temporal resolution in DHS data on gestational length), and overlooked confounders (particularly, longitudinal risk factors on birth outcomes, e.g., gestational hypertension and diabetes) are discussed in previous reports^{10,12,14} and are not repeated here. For instance, the potential bias due to underreporting or misclassifying of stillbirth has been evaluated by a simulation analysis¹², and might cause an underestimated PM_{2.5}-stillbirth association." (line 286-288).

We didn't have direct indicators for the home births or attention for prenatal care, in most of the DHS surveys. As mentioned above, those factors mainly lead to outcome misclassification, which might not be a critical issue to undermine our findings. Additionally, our confidence is increased by controlling for nightlight (i.e., an indicator of developmental level, which can predict the prevalence of home births) between-country heterogeneity in our regression models.

2. How did you account for potentially alternative stillbirth definitions and source types across the various DHS? Was reported antenatal clinic attendance incorporated or tested in your model? E.g. <https://bmjopen.bmj.com/content/12/2/e051675> This would be an important confounder to account for when testing the primary hypothesis.

[Response] Thank for the comment. For stillbirth, as far as we know, studies using DHS surveys, have used different definitions, such as child death at the age of zero. In our study of multiple countries, to guarantee the consistency in definition of stillbirth, we coded the outcome variable according to the specific gestational length for a prematurely terminated pregnancy. We uniformly defined a stillbirth case, if the gestational length ≥ 7 months. To clarity that, in the method section, we add a few words, which read as "The major health outcome for this study was stillbirth, uniformly defined as babies born with no sign of life at ≥ 28 weeks of gestation for all the studied countries. Because the DHS reports gestation in months, here, all early terminated pregnancies with a gestational length ≥ 7 months were defined as stillbirths." The outcome was based on the same source (e.g., the gestational length) in different surveys. We didn't consider any other definition of

stillbirth (e.g., child death at the age of zero) in DHS surveys.

For antenatal clinic attendance, first, we agree with the reviewer that omitting some longitudinal covariates is a key limitation in our study. However, the longitudinal variables, including antenatal clinic attendance, are not available in DHS surveys. DHS data were collected in a cross-sectional way and each mother was visited only once. To clarify that, in the limitation section, we add a few words, which read as “Those related to the development of PM_{2.5}–stillbirth curves, including those on the data quality (e.g., underreporting of stillbirth, misclassified outcomes and recall bias), exposure assessment (e.g., exposure misclassification introduced by residential mobility during pregnancy, and limited temporal resolution in DHS data on gestational length), and **overlooked confounders (particularly, longitudinal risk factors on birth outcomes, e.g., antenatal clinic attendance, gestational hypertension and diabetes)** are discussed in previous reports and are not repeated here.” (line 284-286). Second, one aim of developing the self-matched design is to overcome the weaknesses brought by the omitted longitudinal factors. For instance, the rate of antenatal clinic attendance is usually determined by local medical service, and thus may not be varied dramatically within the same mother. Therefore, without measurement, the variation in antenatal clinic attendance between mothers (e.g., a mother in urban area vs another in rural area) can be controlled for by the design itself. Third, we don’t worry about omitting antenatal clinic attendance, which can be correlated with PM_{2.5} due to their co-determiner, socioeconomic status (SES). After controlling for nightlight, the SES indicator, we believe that the correlation between PM_{2.5} and antenatal clinic attendance can be eliminated or decreased.

3. Fixed effect – genetics (line 170) – unclear what this means and how this was measured/quantified?

[Response] Thank you for the comment. Individual genetic traits are commonly regarded as constant characteristic throughout one’s life. For self-matched study design, the comparisons are performed within the individual, so maternal genetic factors are totally controlled for under the design. In the regression model, the effects of constant or slowly-variant covariates including genetic information are estimated by the fixed-effect term. In DHS surveys, genetic factors were not measured, but were controlled for by the design. Here, we use the genetics as an example to show how the design helps to account for unmeasured confounders. Such confounders also include other factors, e.g., the culture.

4. As is now standard practice re: GATHER compliance – I would suggest including a completed GATHER statement as part of the supplementary materials and also providing a link to a GitHub repository (or similar platform) with all the data processing and modelling scripts.

[Response] Suggestion accepted. We submit the GATHER statement and R codes in the supplementary data. We cannot share the DHS data, which are owned by the third party, through GitHub repository. However, after filling in a brief application, the DHS data can be freely obtained from the official website: <https://dhsprogram.com/>.

REVIEWER COMMENTS

Reviewer #1 (Remarks to the Author):

In this study, authors developed age-specific exposure–response curves based on individual-level data from 54 LMICs to assess PM2.5-related stillbirth risk in 137 countries. They found that 39.7% of stillbirths associated with PM2.5. Compared with previous studies, the sample size of this study is large and many factors are considered, which improves the confidence on estimates to a certain extent. The topic of this study is interesting and can deepens readers' understanding of the health hazards of PM2.5. However, some minor concerns should be improved before publication:

1. Please adjust the font of “ $\mu\text{g}/\text{mL}$ ” in all Figures.
2. Please increase the resolution of Figures S3 and S9.
3. Please adjust the Y-axis (left) spacing in Figure S4.

Dear reviewers,

Thank you for giving us the opportunity to re-submit a revised draft of the manuscript "Estimation of Stillbirths Attributable to Ambient Fine Particles in 137 Countries". We really appreciate the time and effort that you dedicated to improving our paper and are grateful for the valuable comments and suggestions. We have read all comments and adjusted the figures, and the detailed point-by-point responses to the reviewers' concerns and comments are listed as follows. For easy reading, except for the revised version, we also submit a clean version of manuscript.

Reviewer #1 (Remarks to the Author):

In this study, authors developed age-specific exposure–response curves based on individual-level data from 54 LMICs to assess PM_{2.5}-related stillbirth risk in 137 countries. They found that 39.7% of stillbirths associated with PM_{2.5}. Compared with previous studies, the sample size of this study is large and many factors are considered, which improves the confidence on estimates to a certain extent. The topic of this study is interesting and can deepens readers' understanding of the health hazards of PM_{2.5}. However, some minor concerns should be improved before publication:

1. Please adjust the font of “µg/mL” in all Figures.

[Response] Thanks for the suggestion. To best of our knowledge, within the community of environmental science and environmental health, the commonly-used unit for PM_{2.5} is “µg/m³”. Particularly, for the WHO air quality guidelines, the pollutants including PM_{2.5} are measured by “µg/m³”. Therefore, we prefer to use “µg/m³” in this manuscript.

2. Please increase the resolution of Figures S3 and S9.

[Response] Suggestion accepted. We have increased the resolution of Figure S3 and Figure S9.

3. Please adjust the Y-axis (left) spacing in Figure S4.

[Response] Suggestion accepted. We have adjusted the Y-axis (left) spacing in Figure S4.